# Human sperm steer with second harmonics of the flagellar beat

Guglielmo Saggiorato [1,3], Luis Alvarez [3], Jan F. Jikeli[2,4], U. Benjamin Kaupp[3], Gerhard Gompper [1] & Jens Elgeti [1]

Sperm are propelled by bending waves traveling along their flagellum. For steering in gradients of sensory cues, sperm adjust the flagellar waveform. Symmetric and asymmetric waveforms result in straight and curved swimming paths, respectively. Two mechanisms causing spatially asymmetric waveforms have been proposed: an average flagellar curvature and buckling. We image flagella of human sperm tethered with the head to a surface. The waveform is characterized by a fundamental beat frequency and its second harmonic. The superposition of harmonics breaks the beat symmetry temporally rather than spatially. As a result, sperm rotate around the tethering point. The rotation velocity is determined by the second-harmonic amplitude and phase. Stimulation with the female sex hormone progesterone enhances the second-harmonic contribution and, thereby, modulates sperm rotation. Higher beat frequency components exist in other flagellated cells; therefore, this steering mechanism might be widespread and could inspire the design of synthetic microswimmers.

[1] Institute of Complex Systems and Institute for Advanced Simulation, Forschungszentrum Jülich, 52425 Jülich, Germany. [2] Laboratoire de Physique Théorique et Modèles Statistiques, CNRS, Université Paris-Sud, Université Paris-Saclay, 91405 Orsay, France. [3] Department of Molecular Sensory Systems, Center of Advanced European Studies and Research (CAESAR), 53175 Bonn, Germany. [4] Biophysical Imaging, Institute of Innate Immunity, University Hospital Bonn, 53127 Bonn, Germany. Guglielmo Saggiorato and Luis Alvarez contributed equally to this work. Correspondence and requests for materials should be addressed to L.A. (email: luis.alvarez@caesar.de) or to J.E. (email: j.elgeti@fz-juelich.de)

Many microorganisms and cells are propelled by motile flagella or cilia, i.e., hair-like protrusions that extend from the cell surface[1–3]. The beat patterns of flagella vary among cells. The green algae *Chlamydomonas reinhardtii* moves by a breaststroke-like beat of a pair of flagella that alternate between in-phase and anti-phase states of the beat[4]. The bacterium *Escherichia coli* is propelled by the rotary movement of a helical bundle of flagella; when the rotary direction reverts, the flagellar bundle disentangles and the cell randomly adopts a new swimming direction[5]. Most animal sperm swim by means of bending waves traveling from the head to the tip of the flagellum. Near a surface, sperm swim on a curvilinear path[6,7], which is thought to result from the spatial asymmetry of bending waves. Two different mechanisms have been proposed that could generate a flagellar asymmetry: a dynamic buckling instability resulting from flagellar compression by internal forces[8,9] or an average intrinsic curvature[6,10,11].

It has been proposed that the flagellum also serves as antenna that registers sensory cues as diverse as chemoattractant molecules[12–14], fluid flow[15,16], or temperature[17,18]. The sensory cues modify the flagellar beat pattern and, thereby, guide sperm to the egg[19,20]. Whereas chemotaxis, the directed movement in a chemical gradient, has been firmly established in sperm from marine invertebrates and plants[14,20–25], it is debated which sensory cues guide mammalian, in particular human sperm to the egg[12,13,26]. Whichever mechanisms

might underlie steering, it is commonly accepted that changes in intracellular $Ca^{2+}$ concentration ($[Ca^{2+}]_i$) modulate the flagellar beat and, thereby, swimming direction[19,27–30]. However, it is not known whether an intrinsic flagellar curvature, or a buckling instability, or some other mechanism is used for steering and how sensory cues modulate any of these mechanisms.

Here we study the flagellar beat pattern of tethered human sperm by high-speed, high-precision video microscopy and by theoretical and computational analysis. We show that the beat pattern is characterized by a superposition of two bending waves—with a fundamental frequency and its second harmonic—traveling down the flagellum. The second harmonic breaks the symmetry of the overall waveform by a temporal rather than by a spatial mechanism. The sexual hormone progesterone, which evokes a $Ca^{2+}$ influx, enhances the second-harmonic contribution and changes the rotation velocity of the cell. Our analysis suggests a novel temporal steering mechanism of sperm and uniflagellated eukaryotic microswimmers in general.

## Results

**Two harmonics shape the flagellar beat**. We monitored the flagellar beat of human sperm in a shallow recording chamber (150 μm depth) filled with an aqueous buffer solution (viscosity ~0.7 mPa s at 37 °C; Fig. 1a). While swimming near a surface at low viscosities (<20 mPa·s), human sperm undergo a rolling

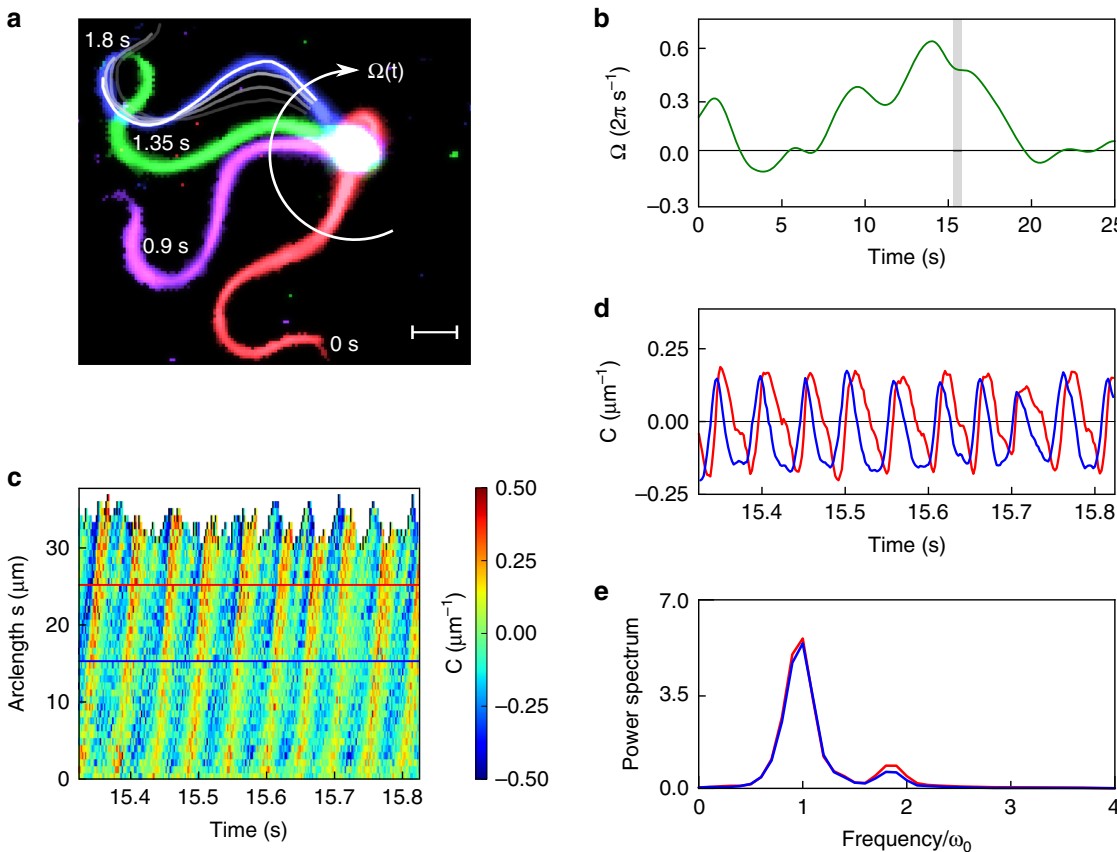

**Fig. 1** The flagellar beat pattern of human sperm displays a second-harmonic component. **a** Four snapshots of a tethered human sperm that rotates clockwise around the tethering point with rotation velocity $\Omega(t)$. Each color corresponds to a different snapshot taken at the indicated time. White and gray lines below the blue snapshot show the tracked flagellum at consecutive frames acquired at 2 ms intervals. Scale bar represents 5 μm. **b** Rotation velocity of the cell around the tethering point. The gray area indicates the time interval that is further analyzed in **c**, **d**. **c** Kymograph of the flagellar curvature during approximately 10 beat cycles (0.5 s). The curvature corresponding to the two horizontal lines is plotted in **d**. **d** Curvature of the flagellum at segments located at 15 μm (blue) and 25 μm (red) from the head. The curvature displays a sawtooth-like profile. **e** Power spectrum of the curvature at 15 μm and 25 μm. The fundamental frequency is $\omega_o = 20$ Hz

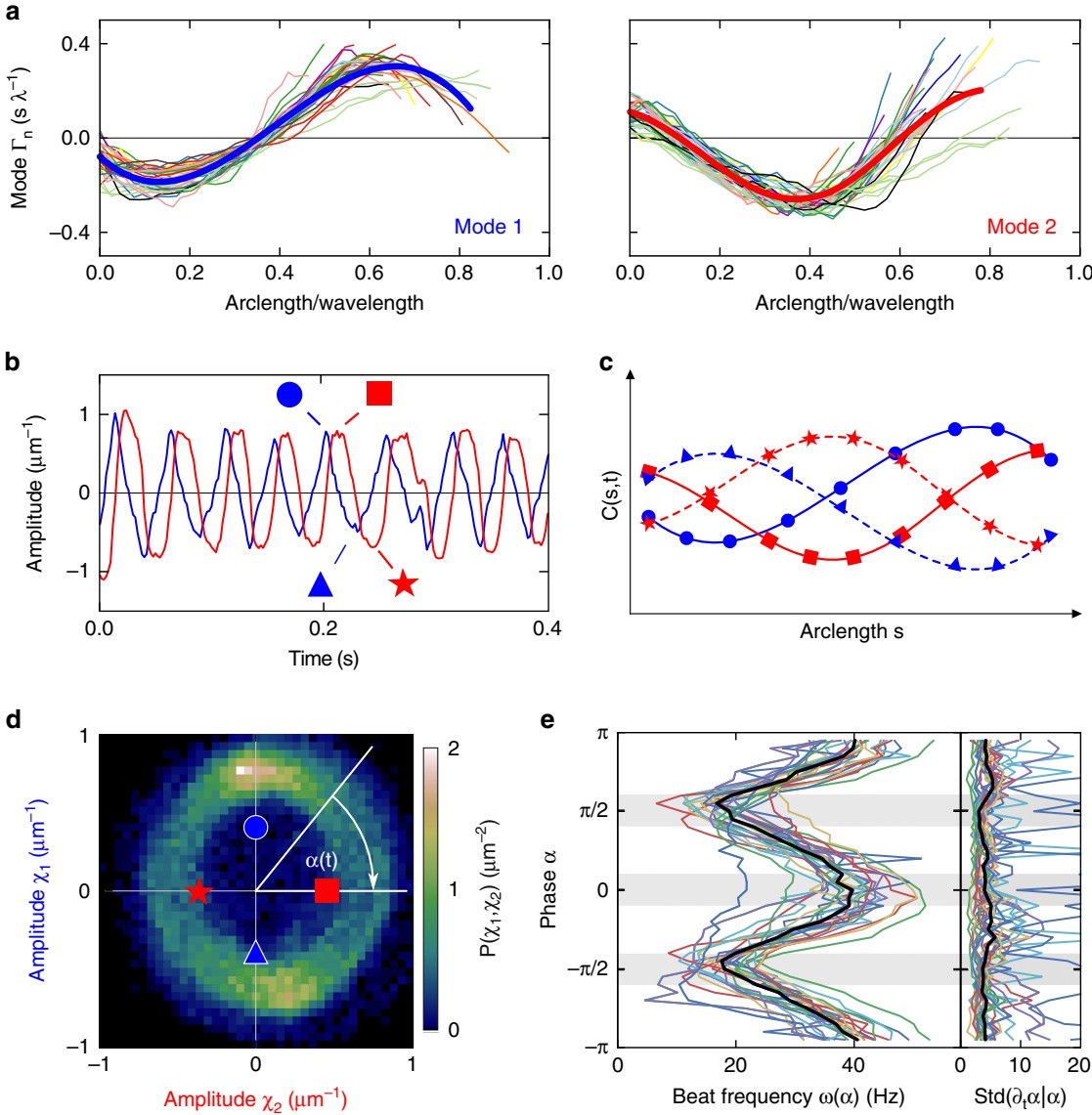

**Fig. 2** Principal-component analysis of the flagellar beat. **a** Superposition of the two main normal modes of the flagellar beat for $n = 35$ human sperm cells. Each trace corresponds to a different cell. Although each cell has a different set of eigenmodes, when rescaled by the wavelength $\lambda$, the individual modes collapse onto a common curve (solid blue and red lines; see Methods). The wavelength appears longer than the flagellum, because it is traced only partially. **b** Time evolution of the mode amplitudes $\chi_1$ and $\chi_2$ for a representative recording. Of note, $\chi_2$ lags behind $\chi_1$ by a phase of $\pi/2$. The symbols (blue circles, red squares, blue triangles, red stars) indicate the modes shown in panel **c**, and correspond to the beat phase depicted in **d**. **c** Illustration of the composition of the beat by principal modes. The peak of the wave travels from left to right. **d** Histogram of the joint probability $P(\chi_1, \chi_2)$, averaged for 1.5 s from 26 different cells. During a full beat cycle, the phase $\alpha = \arctan(\chi_2/\chi_1)$ varies between 0 and $2\pi$. The symbols indicate the phase of the amplitudes in **b**. **e** Beat frequency at a fixed phase for each cell as $\omega(\alpha) = \langle \partial_t \alpha | \alpha \rangle$. The gray stripes highlight the phases of minimal and maximal frequency (20 Hz and 40 Hz, respectively). The black line indicates the median. The standard deviation of $\omega(\alpha)$ is nearly constant (right panel)

motion[8,31]. Rolling occurs despite the fact that the flagellar beat is almost planar[8]. We prevent cell rolling by tethering sperm with their head to the recording chamber. Under these conditions, the beat plane remains parallel to the surface, which facilitates tracking and imaging of the flagellar motion. Sperm revolve around the tethered head with a rotation velocity $\Omega$ that varied smoothly over time between 0 and 0.5 Hz (Fig. 1a, b). The flagellar shape, extracted by image processing, was characterized by the local curvature $C(s,t)$ at time $t$ and arclength $s$ along the flagellum. The curvature profile shows the well-known bending wave propagating along the flagellum from the midpiece to the tip (Fig. 1c). However, at a fixed arclength position, the curvature deviates in time from a perfect sinusoidal wave;

instead, the curvature displays an asymmetric sawtooth-like profile in time, suggesting that multiple beat frequencies contribute to the overall waveform (Fig. 1d). Fourier analysis of the beat pattern reveals a fundamental beat frequency $\omega_o$ of about 20 Hz, but also higher-frequency components, mainly the second harmonic (Fig. 1e).

Principal component analysis[32] provides further support for the presence of a second harmonic (Fig. 2). We decomposed the curvature profile $C(s,t)$ into normal modes $\Gamma_n(s)$ (Fig. 2a):

$$C(s,t) = \sum_n \chi_n(t) \Gamma_n(s), \quad (1)$$

where $\chi_n(t)$ is the amplitude of the n-th mode (Fig. 2b, c). The curvature is sufficiently well described using the first two modes (Fig. 2a), that account for about 90% of the signal (35 cells; Methods section and Supplementary Fig. 2). The beat wavelength varied among sperm. However, after rescaling the arclength by the beat wavelength, the first two modes from different cells can be superimposed (Fig. 2a). Thus, the superposition of two eigenmodes recapitulates fairly well the beat pattern of human sperm.

The mode amplitudes $\chi_{1,2}(t)$ encode the time dependence of the flagellar shape. The probability $P(\chi_1,\chi_2)$ of observing a particular combination of mode amplitudes $\chi_1$ and $\chi_2$ adopts a typical limit-cycle shape (Fig. 2d)[32–34]. For a single-frequency cycle, the probability density would adopt an isotropic ring-like distribution, characterized by $(\chi_1,\chi_2) = a(\cos(\omega_o t), \sin(\omega_o t))$. However, the measured probability density displays two regions with higher probabilities at the "north" and "south" poles of the beat cycle, indicating the presence of a second harmonic (Fig. 2d). The average phase velocity for a given phase, $\langle \partial_t \alpha | \alpha \rangle$, reveals that the frequency smoothly oscillates between about 20 Hz and 40 Hz during each beat cycle (Fig. 2e). Thus, the beat pattern is indeed characterized by a fundamental frequency and its second harmonic.

**A second-harmonic mode breaks the mirror symmetry temporally**. A planar beat with a single frequency becomes its mirror image after half a beat period $\tau/2$ (i.e. $C(s,t) = -C(s,t + \tau/2)$). Sperm using such a mirror-symmetric flagellar beat would swim on a straight path, and tethered sperm would not rotate. The second harmonic breaks this mirror symmetry; consequently, curved swimming paths and tethered sperm rotation becomes possible. We examined theoretically how this broken symmetry generates the torque that drives rotation. With the "small-amplitude approximation"[2,35], the waveform of a flagellum oriented on average parallel to the x-axis can be described by a superposition of first and second harmonics:

$$y(x, t) = y_1\sin(kx - \omega_o t) + y_2\sin(kx - 2\omega_o t + \phi), \quad (2)$$

where $k$ is the wave vector, $\phi$ is the phase shift between the two modes, and $\omega_o$ is the fundamental frequency. Note that Eq. (2) can be rewritten as $y(x,t) = Y(t)\sin(kx - \omega_o t + \Phi(t))$, where the amplitude $Y$ and phase $\Phi$ are functions of time. At any instant in time $t_o$, the shape $y(x,t_o)$ is still a sine wave, and is thus mirror symmetric in space. Therefore, no average flagellar curvature is produced. By contrast, because the amplitude $Y(t)$ is time-modulated, at any given point $x_o$ the temporal dependence of $y(x_o,t)$ is asymmetric in time. Such an asymmetric beat pattern allows steering, because the hydrodynamic drag forces during the two halves of the beat cycle do not cancel. For amplitude ratios $y_2/y_1 \lesssim 0.3$, $y(x_o,t)$ resembles an asymmetric sawtooth-like profile as in Fig. 2b.

The hydrodynamic drag on the flagellum is anisotropic, i.e. the drag coefficients in the perpendicular ($\xi_\perp$) and tangential directions ($\xi_\parallel$) are not equal. Each point along the flagellum is subjected to the drag force $\mathbf{f}(x,t) = -\xi_\perp \mathbf{v}_\perp - \xi_\parallel \mathbf{v}_\parallel$, where $\mathbf{v}(x,t) = (0, \partial_t y)$ is the velocity of the filament at time $t$ and position $x$[2,7,36,37]. The net perpendicular force, averaged over one beat cycle is (see Supplementary Note)

$$f_y(x) = \omega_o k^2 \left( \xi_\perp - \xi_\parallel \right) y_1^2 y_2 \cos(kx - \phi) \quad (3)$$

In the presence of a second harmonic ($y_2 \neq 0$), the force $f_y$, integrated over the whole flagellum, does not vanish. The rotation velocity $\Omega$ around the tethering point is obtained by torque balance: the torque generated around the tethering point $T_a \approx \int_0^L x f_y(x) \cdot dx$ equals the viscous torque. For comparison with

experiments, it is more useful to describe the waveform in terms of local curvature $C(s,t)$, with amplitudes $C_1$ and $C_2$ instead of $y(x,t)$ with amplitudes $y_1$ and $y_2$ in Eq. (2). In addition, a small-curvature approximation is more accurate for larger beat amplitudes. The general result for $\Omega$ (Supplementary Note) depends on wavelength $\lambda$ and flagellum length $L$; for $\lambda \to L$, a simple relation is obtained:

$$\Omega = -\omega_o \frac{3L^3}{4(2\pi)^4} \frac{\xi_\perp - \xi_\parallel}{\xi_\perp} C_1^2 C_2 \sin \phi. \quad (4)$$

Eq. (4) illustrates that rotation results from the superposition of the first and the second harmonics coupled to the anisotropic drag in a similar way as Eq. (3). Note that $\Omega$ depends both on the amplitude $C_2$ of the second harmonic and on the phase shift $\phi$ between the two modes. We refer to $C_2 \sin(\phi)$ as the "second-harmonic intensity".

**Second-harmonic intensity and rotation velocity correlate**. The experimentally observed rotation velocity slowly varies with time (Fig. 1b), providing the means to test the predictions from Eq. (4). We determined the phase ($\phi$) and amplitude ($C_2$) of the second harmonic from the spectrogram of the flagellar curvature (Methods and Supplementary Fig. 3) and compared the second-harmonic intensity with the rotation velocity $\Omega$ (Fig. 3a). For each cell ($n = 35$), the correlation coefficient $R$ between the normalized rotation velocity $\Omega(t)/\omega_o$ and second harmonic intensity $C_2(t)\sin(\phi(t))$ was calculated by time averaging over the course of the experiment. To account for the approximations introduced for the derivation of Eq. (4), the phase $\phi(t)$ is corrected by a constant phase shift $\phi_o$ to yield $\phi_{eff}(t) = \phi(t) + \phi_o$. The constant shift is chosen such as to maximize the correlation coefficient $R$. We find that the second-harmonic intensity and the rotation velocity are highly correlated (Fig. 3a–c) ($R = 0.91 \pm 0.13$, mean $\pm$ SD).

Alternatively, an average intrinsic curvature ($C_o$) of the flagellum might contribute to the rotation. An intrinsic curvature, which can generate an asymmetric beat, has been observed for some cilia and flagella[6,10,11]. The small-curvature calculation (Methods and Supplementary Note) predicts that

$$\Omega = \omega_o \frac{L^3}{(2\pi)^6} \frac{\xi_\perp - \xi_\parallel}{\xi_\perp} C_1^2 C_0 \cdot \pi(\pi^2 - 3). \quad (5)$$

Thus, for equal magnitudes of $C_o$ and $C_2$, both mechanisms contribute alike to the rotation frequency (compare Eqs. (4) and (5)). However, the average intrinsic curvature of the flagellum is usually much smaller than the amplitude of the second harmonic ($|C_o|/|C_2| = 0.13$; Supplementary Note and Supplementary Fig. 5). Therefore, we conclude that the second-harmonic contribution to rotation velocity dominates. Accordingly, we find that the correlation of $C_o$ with the rotation frequency is weak ($R = 0.13 \pm 0.65$; $n = 35$, mean $\pm$ SD). However, sometimes the average curvature, second harmonic intensity, and rotation velocity display a similar time course (Fig. 3a). In summary, these results support the hypothesis that human sperm steer with the second harmonic.

**Changes in [Ca²⁺] control second-harmonic intensity**. The flagellar beat of sperm and the steering response is controlled by changes in $[Ca^{2+}]_i$[38]. The female sex hormone progesterone evokes robust $Ca^{2+}$ entry into human sperm by activating the CatSper $Ca^{2+}$ channel[39,40]. The ensuing change in the flagellar beat pattern has been proposed to underlie hyperactivated motility and chemotaxis[13,41–43]. We used progesterone stimulation to examine whether $Ca^{2+}$ modulates the second-harmonic contribution. Sperm were imaged before and after photo-release of progesterone from a caged derivative (Fig. 3d)[27]. In Fig. 3e, f, we compare the beat pattern during 0.5 s before and after the

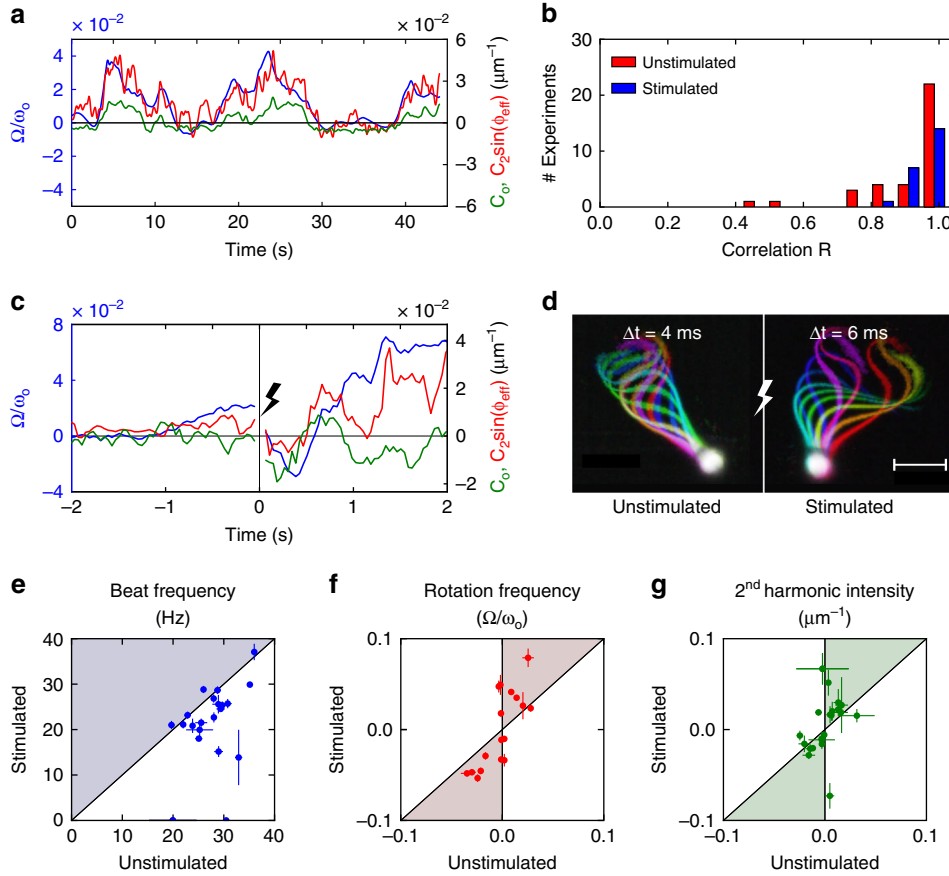

**Fig. 3** Second-harmonic intensity correlates with rotation velocity and is enhanced by progesterone. **a** Normalized rotation velocity (blue line), second-harmonic intensity (red line), and average curvature (green line) for a representative sperm cell. **b** Histogram of the correlation $R(\Omega/\omega_o, C_2 \sin(\phi_{eff}))$. Red bars refer to unstimulated human sperm, blue bars to sperm stimulated with progesterone. We never observed anti-correlation ($R < 0$). **c** Normalized rotation velocity (blue line) and second-harmonic intensity (red line), and average curvature (green line) 2.0 s before and after the release of progesterone with a flash of UV light (at $t = 0$). **d** Stroboscopic views of a sperm cell before (left) and after (right) stimulation with progesterone. Flagellar snapshots were recorded at $\Delta t = 4$ ms (left) and $\Delta t = 6$ ms (right) intervals. Scale bar represents 10 μm. **e–g** Cell by cell comparison of the beat frequency (**e**), the rotation frequency (**f**), and the second-harmonic intensity (**g**), before and after progesterone release. Average values during 0.5 s before and after the stimulus. Points inside the colored areas correspond to an increase after the stimulation. Error bars are SD

release of progesterone. Although progesterone slowed down the beat frequency of human sperm (Fig. 3e), the rotation around the tethering point was enhanced (Fig. 3f). A direct comparison of the second-harmonic contribution before and after the release (Fig. 3c, g) demonstrates that progesterone modulates the second-harmonic intensity and, thereby, the rotation velocity; moreover, both measures are highly correlated (Fig. 3b, c, f, g). The strong second-harmonic component might thus represent the mechanism of hyperactivated beating of human sperm upon progesterone stimulation.

**Sperm can navigate by adjusting the phase between harmonics**. Beyond a purely geometric description of the shape, we study by simulation the elasticity, forces, and the power generated or dissipated during a flagellar beat. A sperm cell is modeled as an actively beating filament of bending rigidity κ; local hydro-dynamic interactions resulting from the dynamic coupling between different portions of the flagellum via the induced fluid flow are taken into account by anisotropic drag[6,35,37]. The filament is driven by active bending torques $T(s, t)$, assuming a superposition of two traveling waves,

$$T(s,t) = T_1 \sin(ks - \omega_o t) + T_2 \sin(ks - 2\omega_o t + \psi). \quad (6)$$

Due to hydrodynamic and boundary effects, the phase shift ψ of the torque can be different from the phase shift ϕ of the flagellar curvature in Eq. (4). All parameters in Eq. (6) and the bending rigidity κ were derived by fitting to experimental data, including flagellar waveform, rotation velocity, and normal modes (Supplementary Note). The simulation, which reproduces the beat pattern reasonably well (Fig. 4a, Supplementary Movie 1), provides several insights. First, constant torque amplitudes $T_1$ and $T_2$ along the flagellar arclength suffice to account for the experimentally observed beat shapes, including the very high curvature of the end piece. Thus, no structural inhomogeneity or differential motor activity along the flagellum is needed to account for this peculiarity of the flagellar beat shape. Second, although the bending forces are mirror-symmetric with respect to the filament displacement, a small average curvature is generated by the superposition of two harmonics that breaks the mirror symmetry of the beat waveform in both time (second harmonic) and in space (average curvature) (Fig. 4b, d). Third, simulations confirm two predictions from the small-curvature approximation Eq. (4): The rotation velocity Ω scales linearly both with $T_2$ (Fig. 4c, d) and with the sine of the phase ψ (Fig. 4d, Supplementary Fig. 7). Fourth, for wavelength λ < L, the rotation velocity is largely independent of the wavelength; however, for longer wavelengths, the rotation velocity decreases (Fig. 4c).

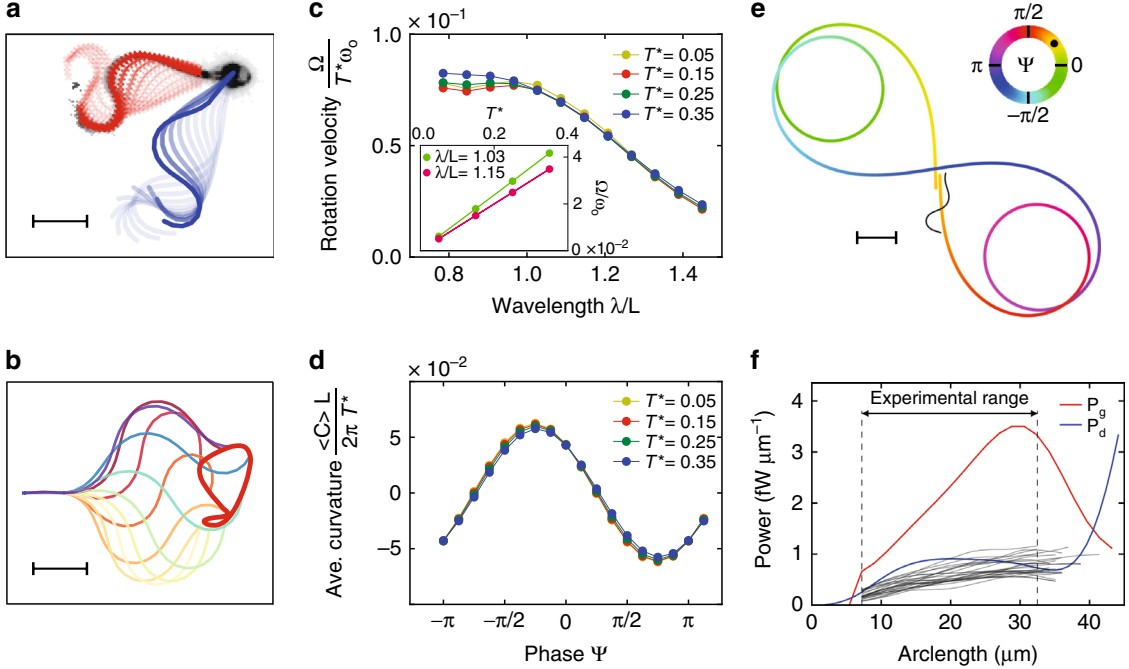

**Fig. 4** Simulations reproduce the beat and steering dynamics. **a** Stroboscopic view of experimental (red) and simulated (blue) beat pattern using an active semi-flexible filament and anisotropic drag force. Time interval between snapshots (fading lines) is $\Delta t = 2$ ms. Simulation parameters: $\kappa = 1.9$ nN μm$^2$, $T_1 \sim 0.65$ nN μm, $T_2 = 0.15 T_1$, $\psi = 2.26$, $\omega_o = 30$ Hz, $L = 41$ μm, $\xi_\perp / \xi_\parallel = 1.81$, $\xi_\parallel = 0.69$ fNs μm$^{-2}$ and $\lambda/L = 0.65$ (Supplementary Movie 1). Scale bar represents 10 μm. **b** Representative simulation of flagellar beat with a second-harmonic amplitude $T_2 = 0.3 T_1$. The mid-piece is aligned for visualization. The time interval between snapshots is $\Delta t = 4$ ms. The red thick line shows the non-symmetric trajectory of the flagellum tip. Scale bar represents 10 μm. **c** Rotation velocity $\Omega$ vs. normalized wavelength $\lambda$. Note that $\Omega$ has been normalized to the second-harmonic torque amplitude. The inset shows that $\Omega$ scales linearly with $T^* = T_2/T_1$. **d** Average curvature $<C>$ vs. phase $\psi$ of the second-harmonic torque. Note that the curvature has been normalized by $T^* = T_2/T_1$. **e** Simulated sperm trajectory resulting from a slowly changing phase $\psi$ over time (phase indicated by the color of the trajectory). By modulating the phase, sperm swim on curvilinear paths (Supplementary Movie 2). Scale bar represents 20 μm. **f** Average dissipated power $P_d$ (blue) vs. generated power $P_g$ (red) in simulations, and average dissipated power measured in experiments (gray lines). The simulated dissipated power shows good agreement with the experimental results. Of note, power is relocated along the flagellum

Finally, simulations of freely swimming sperm in 2D show that the curvature of the swimming path is controlled by the phase $\psi$ (Fig. 4e), i.e. sperm could navigate by adjusting the phase $\psi$ between the two harmonics.

**Energy consumption and dissipation.** Several aspects regarding the energetics of motile cilia and flagella have been studied, including traveling waves, power-and-recovery stroke, and metachronal waves[33,44–49]. For propulsion, not all beating gaits offer the same efficiency of energy consumption[50–52]. In fact, the flagellar beat pattern can be predicted from optimal swimming efficiency[51]. However, quantitative estimates of how power is used for bending and how power is dissipated along the flagellum of microswimmers are lacking[53–55].

Our simulations provide insight into the energetics of beating. Comparison of experimental results with simulations is only possible for power $P_d$ dissipated due to drag forces. The dissipated power $P_d$ (Supplementary Note) increases along the flagellum within about 10 μm from the head, stays roughly constant for 25 μm along the entire principal piece, and then steeply rises towards the end piece, where the flagellum moves faster (Fig. 4f). The experimental recordings, which are restricted to a flagellar section between 7 and 35 μm from the head center, agree reasonably well with the simulations (Fig. 4f). The dissipated power remains unaltered after stimulation (Supplementary Fig. 8). From the simulations, we can also estimate the power $P_g$ generated by the instantaneous local torques described by Eq. (6) (Supplementary Note). The generated power $P_g$ increases steadily, becomes

maximal at about 30 μm down the flagellum, and decreases again towards the tip region. The distributions of generated and dissipated power differ along the flagellum: During the steady increase of $P_g$ to its maximum, the dissipated power $P_d$ stays almost constant; then $P_g$ quickly drops, whereas $P_d$ steeply increases thereafter (Fig. 4f). We conclude that the effects of local torques add up in order to generate large beating amplitudes and velocities in the tip region; by contrast amplitudes and velocities are smaller in the mid-piece region due to the tethering constraint and the head drag. Thus, although bending forces in eukaryotic flagella are locally generated along the length of the axoneme, power dissipation due to fluid drag is not equally distributed, yet relocated towards the tip region of the flagellum.

## Discussion

Two spatial mechanisms that can produce asymmetric waveforms of the ciliary or flagellar beat have been proposed previously: average intrinsic curvature or buckling instabilities. Here we identify a novel mechanism of symmetry breaking that is temporal rather than spatial: two traveling waves of fundamental and second-harmonic frequency determine the beat asymmetry by their phase relation and the relative amplitude of each component.

Symmetry breaking by an intrinsic curvature can emerge from structural features such as the central apparatus of the axoneme, elastic filaments such as nexin links, or dynein motors whose properties vary along the circumference or the long axis of the axoneme[56–58]. By contrast, the second harmonic mechanism does

not require spatial or structural inhomogeneity. Consistent with this idea and experiment, we find that simulations using a constant torque amplitude for the fundamental and second-harmonic components along the flagellar arclength (Eq. 6, Fig. 4a, Supplementary Movie 1), to produce flagellar beat shapes with varying curvature along the flagellum.

Hydrodynamic simulations suggest that intrinsic curvature of the midpiece affects swimming path curvature[10]. However, a spatial intrinsic curvature and a temporal second-harmonic component are not mutually exclusive. In fact, Fourier analysis of beat waveforms from different sperm species and Chlamydomonas[11] reveals a zero component or intrinsic curvature component[6] and at least two other components: a principal and a second harmonic[6,8]. However, in human sperm, the intrinsic curvature is small (Fig. 3a, c, Supplementary Fig. 5) and its modulation was not favored as a steering mechanism during rheotaxis[8]. Alternatively, buckling instabilities have been proposed to produce asymmetric beating[8,9]. At higher frequencies, these instabilities are enhanced due to higher shear forces and flagellar compression[9]. However, progesterone stimulation slows down the beat frequency considerably, which is expected to lower shear forces and reduce buckling[9]. Assuming that buckling underlies asymmetric beating, reducing buckling amplitude is expected to reduce rotation velocity. By contrast, the rotation frequency and second-harmonic intensity are enhanced by progesterone stimulation (Fig. 3e–g). These experiments, therefore, are difficult to reconcile with dynamic buckling instabilities underlying mirror symmetry breaking in human sperm. Additional experimental studies are required to elucidate the role of dynamic buckling or flutter instabilities for shaping the beat waveform. In fact, sperm feature different morphologies and swim in vastly different environments[23,26]. Thus, we envisage that future work will uncover a panoply of swimming modes produced by second harmonics, static components, and buckling instabilities.

It might be argued that the second harmonics of the beat result from the physical constraints imposed by cell tethering. However, second harmonics have also been noted in freely swimming sperm from bull[6], and in reactivated flagella from Chlamydomonas[59]. Moreover, simulations of the flagellar beat show that the second-harmonic intensity can control both the path curvature of freely swimming sperm and the rotation velocity of tethered sperm (Fig. 4e, Supplementary Movie 2). Together, these observations indicate that the second-harmonic mechanism per se does not result from tethering constraints.

For steering, sperm might modulate the phase between the two harmonics. If the phase were constant, sperm would swim on a circular path with no net drift. For a net drift to occur, sperm must adjust the phase between the two harmonics in time. Furthermore, we find that progesterone-evoked $Ca^{2+}$ influx enhances the relative contribution of the second harmonic to the overall beat. Thus, the dynamics of principal and second-harmonic traveling waves could steer sperm across gradients of sensory cues that modulate the $Ca^{2+}$ concentration. Steering by a second-harmonic mechanism should be also valid for rolling sperm moving freely in three dimensions (3D). The 3D swimming path resulting from a non-planar periodic beat is a helix, or a twisted ribbon for a helix radius near zero[21,60–62]. Analogous to the 2D scenario, for steering in 3D the phase between the two harmonics needs to be adjusted as well. Thus, the proposed second-harmonic mechanism might be also valid for 3D swimming. However, studies of human sperm swimming in 3D reveal that only a small fraction of sperm (4–5%) swims on a regular helical path[63]. Future work needs to stablish the role of the second harmonic for steering in 3D when the swimming path deviates from a perfect helix. As an aside, higher harmonics have been suggested to control 3D steering during phototaxis of Chlamydomonas[64].

The mechanisms underlying the second harmonic and its modulation by $Ca^{2+}$ are not known. Here we discuss some potential mechanisms. Second harmonic generation might involve dynein motors. Dynein arms behave as endogenous oscillators that slide microtubules with a frequency set by the ATP concentration[65]. In fact, isolated Chlamydomonas flagella that were reactivated with different ATP concentrations display beat amplitudes with two peak resonances at 30 and 60 Hz[59]. For a given ATP concentration, different dynein isoforms translocate microtubules at different velocities[66]. Thus, fundamental and second harmonics could be produced by different motor subsets. A novel model for the generation of flagellar waves[67] predicts that a combination of steady forces, i.e. dynein activity, and dynamic instability can create oscillatory waves propagating down the flagellum. The beat frequency does not rely on switching or modulation of motor activity, but on the force density produced by motors on the flagellum. In fact, detergent-extracted axonemes from sea urchin sperm and flagella from Chlamydomonas mutants lacking the outer dynein arms beat at about half the frequency[68–70], supporting the notion that a lower motor activity or density results in reduced beat frequency[67]. By contrast, Chlamydomonas flagella with defective inner dynein arms beat at about the same frequency as wild type flagella, suggesting that inner and outer dynein arms produce different beat frequencies[69,70]. Finally, two different sources of flagellar nonlinearities are possible: structural nonlinearities resulting from anharmonic mechanical properties of complex materials and geometric nonlinearities resulting from body deformation. Nonlinearities could also produce a spectrum of flagellar beat frequencies, particularly first and second harmonics[67]. We investigated this hypothesis by simulations of elastic filaments including geometric nonlinearities and a bending torque with a fundamental component superimposed upon an average torque, but no second harmonic component (Supplementary Information). Indeed, nonlinearities result in a spontaneous second-harmonic frequency. However, a bending torque with fundamental and second-harmonic components (Eq. (6)) clearly provides the best fit to the experimental results. Future studies need to address the mechanisms by which a second-harmonic mode is created and tuned for steering.

## Methods

**Sperm preparation.** Samples of human semen were from healthy donors with their consent. Sperm were purified by a "swim-up" procedure in human tubal fluid (HTF) containing (in mM): 97.8 NaCl, 4.69 KCl, 0.2 $MgSO_4$, 0.37 $KH_2PO_4$, 2.04 $CaCl_2$, 0.33 Na-pyruvate, 21.4 lactic acid, 2.78 glucose, 21 HEPES, and 4 $NaHCO_3$; pH was adjusted between 7.3 and 7.4 with NaOH. The viscosity of HTF at 37 °C (0.73 ± 0.01 mPa.s) was measured using a rheometer (Haake RheoStress600; Thermo Electron Corporation). After washing, human serum albumin (HSA, Scientific Irvine, USA; 3 mg ml$^{-1}$) was added, and sperm were incubated for at least 1 h at 37 °C and 10% $CO_2$ atmosphere.

**Sperm motility.** Single sperm cells were imaged in custom-made observation chambers of 150 μm depth. To gently tether the head of sperm cells to the glass surface, the HSA concentration in the buffer was reduced to 1 μg ml$^{-1}$, resulting in a large fraction of cells tethered to the surface with the head, but the flagellum was freely beating. The flagellar beat was recorded under an inverted microscope (IX71; Olympus) equipped with a dark-field condenser, a 20x objective (UPLANFL; NA 0.5), and additional 1.6x magnification lenses (32× final magnification). The temperature of the microscope was adjusted to 37 °C using an incubator (Life Imaging Services). Illumination was achieved using a red LED (M660L3-C1; Thorlabs), and a custom-made power supply. Images were collected at 500 frames per second using a high-speed CMOS camera (Dimax HD; PCO). For release of progesterone from its caged derivative (1 μM)[27] a brief flash (100 ms) of UV light was used (365-nm LED; M365L2-C; Thorlabs). UV light reached the sample through the backport of the microscope and a 380 nm long-pass dichroic filter (380 DCLP; Chroma). Tracking of the flagellum was achieved with custom-made programs written in MATLAB (Mathworks). The program identified the best threshold for binarization of the image by iteratively reducing the threshold until the expected cell area and coarse flagellar length in the image was achieved. This

was followed by a skeleton operation to identify the flagellum. The position of the head was determined by fitting an ellipse around the tethering point.

**Rotation velocity**. The rotation velocity $\Omega(t)$ is obtained as the time derivative of the angle $\alpha$ between the x-axis and the vector connecting the head tethering point with the first tracked flagellar point. The angle $\alpha$ was filtered with a Gaussian of width 1 s to remove oscillations due to the fast beat.

**Flagellar curvature**. An arclength $s$ sampled every $\Delta s = 0.9\ \mu m$ from head to tip, is assigned to the tracked flagellum. Because the number of tracked flagellar points can differ from frame to frame (compare panels a and b in Supplementary Fig. 1), we analyze only the part of the flagellum that has been tracked for all frames. Arclength is measured from the first tracking point. The curvature $C(s,t)$ is computed as the inverse of the radius of the circle that connects three contiguous points of the tracked flagellum, and is positive (negative) for counter-clockwise (clockwise) bends.

**Principal-component analysis**. The curvature $C(s,t)$ is a matrix of about 30 (in arclength) by 10,000 (in time) entries (Supplementary Fig. 1). We reduce the dimensionality of the dataset by principal-component analysis[32] to filter out white noise. The normal modes $\Gamma_n$ of the curvature are the eigenvectors of the non-standardized covariance matrix $M(s,s') = \langle C(s,t)C(s',t)\rangle_t$, with average curvature $\langle C(s,t)\rangle_t \approx 0$. Modes are sorted according to their eigenvalues $\sigma_n$. The number $n$ of eigenvalues $\sigma_n$ and eigenvectors $\Gamma_n(s)$ is equal to the number of tracking points ($n \approx 30$). We distinguish the relevant modes by comparing the eigenvalues of $M(s,s')$ with the eigenvalues of the correlation matrix of a random curvature $C_{ran} = \eta(s,t)\cdot\sigma$, where $\langle \eta(s,t)\eta(s',t)\rangle = \langle \delta(t-t')\delta(s-s')\rangle$ and $\sigma^2$ is the variance of the original curvature. Only the first three eigenmodes are statistically significant (Supplementary Fig. 2) and contribute with about 95% to the signal. The curvature $C(s,t) = \sum_{i=1}^{3} \Gamma_i(s)\chi_i(t)$ reconstructed from the first three modes and the corresponding amplitudes $\chi_i(t) = \int_0^L C(s,t)\Gamma_i(s)ds$ describes fairly well the experimental data (compare Supplementary Fig. 1a, b), and was used for further analysis.

To identify whether the same modes underlie all observed beat patterns (Fig. 2), we rotate and mirror the first two eigenmodes, $\Gamma_1$ and $\Gamma_2$, of each experiment to maximize similarity with a reference pair of modes, $\Gamma_1^*$ and $\Gamma_2^*$. Because arclength is measured in wavelength units $\Delta s \rightarrow \Delta s/\lambda$, modes are interpolated to correct for the different tracking point density in the scaled representation. To avoid potential artefacts due to rotation and mirroring of modes, we use untransformed modes for other analyses.

**Curvature spectrogram**. We perform a discrete Fourier transform of the curvature $C(t, s = s_o)$ every 30 frames in the time window $(t_o − W/2, t_o + W/2)$ at fixed arclength $s_o$ and time window width $W = 250$ frames. $W$ is a compromise between time and frequency resolution. The peak at the fundamental frequency $\omega_o$ is clearly identified (Supplementary Fig. 3). The second-harmonic amplitude $C_2$ and phase $\phi$ are measured at $2\omega_o$.

**Correlation coefficient R**. The correlation coefficient $R$ between the rotation velocity and second-harmonic intensity is defined as

$$R = \frac{\langle \Omega(t)/\omega_o \cdot C_2(t)\sin(\phi(t) + \phi_o)\rangle_t}{\sqrt{\langle (\Omega(t)/\omega_o)^2\rangle_t \left\langle \left(C_2^2(t)\sin^2(\phi(t) + \phi_o)\right)^2\right\rangle_t}}, \qquad (7)$$

where $\phi(t)$ is obtained directly by Fast-Fourier Transform of $C(t, s = s_o)$ (see Fig. 1c). The phase offset $\phi_o$ is chosen such as to maximize $R$. Note that this is just one constant offset for each experiment with about 1000 data points. We test whether the value of $\phi_o$ is independent of $s_o$ by comparing $R_{25}$, estimated from the curvature at 25 μm, with $R_{15}$, estimated from the curvature at 15 μm, using the phase estimated for $R_{25}$. The results are virtually identical (Supplementary Fig. 4). The correlation coefficient is centered around the values $C_2(t)\sin(\phi(t) + \phi_o)=0$ and $\Omega(t)/\omega_o(t) = 0$. This choice agrees with the expectation that in the absence of a second harmonic the rotation velocity is zero.

**Data availability**. High-speed video recordings, extracted data, and analysis supporting the findings of this study are available in "zenodo" with the identifier doi: 10.5281/zenodo.884626.

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

# ARTICLE

34. Ma, R., Klindt, G. S., Riedel-Kruse, I. H., Jülicher, F. & Friedrich, B. M. Active phase and amplitude fluctuations of flagellar beating. *Phys. Rev. Lett.* **113**, 048101 (2014).
35. Gray, J. & Hancock, G. J. The propulsion of sea-urchin spermatozoa. *J. Exp. Biol.* **32**, 802–814 (1955).
36. Cox, R. G. The motion of long slender bodies in a viscous fluid Part 1. General theory. *J. Fluid Mech.* **44**, 791–810 (1970).
37. Lighthill, J. Flagellar hydrodynamics—Neumann, JV Lecture, 1975. *Siam. Rev.* **18**, 161–230 (1976).
38. Kaupp, U. B. & Alvarez, L. Sperm as microswimmers—navigation and sensing at the physical limit. *Eur. Phys. J. Spec. Top.* **225**, 2119–2139 (2016).
39. Strünker, T. et al. The CatSper channel mediates progesterone-induced $Ca^{2+}$ influx in human sperm. *Nature* **471**, 382–386 (2011).
40. Lishko, P. V., Botchkina, I. L. & Kirichok, Y. Progesterone activates the principal $Ca^{2+}$ channel of human sperm. *Nature* **471**, 387–391 (2011).
41. Harper, C. V., Barratt, C. L. R. & Publicover, S. J. Stimulation of human spermatozoa with progesterone gradients to simulate approach to the oocyte: Induction of $[Ca^{2+}]_i$ oscillations and cyclical transitions in flagellar beating. *J. Biol. Chem.* **279**, 46315–46325 (2004).
42. Armon, L. & Eisenbach, M. Behavioral mechanism during human sperm chemotaxis: involvement of hyperactivation. *PLoS ONE* **6**, e28359 (2011).
43. Suarez, S. S. Control of hyperactivation in sperm. *Hum. Reprod. Update* **14**, 647–657 (2008).
44. Fauci, L. J. & Dillon, R. Biofluidmechanics of reproduction. *Annu. Rev. Fluid Mech.* **38**, 371–394 (2006).
45. Brokaw, C. J. Bend propagation along flagella. *Nature* **209**, 161–163 (1966).
46. Camalet, S. & Jülicher, F. Generic aspects of axonemal beating. *N. J. Phys.* **2**, 241–2423 (2000).
47. Bayly, P. V. & Wilson, K. S. Equations of interdoublet separation during flagella motion reveal mechanisms of wave propagation and instability. *Biophys. J.* **107**, 1756–1772 (2014).
48. Vilfan, A. & Jülicher, F. Hydrodynamic flow patterns and synchronization of beating cilia. *Phys. Rev. Lett.* **96**, 058102 (2006).
49. Elgeti, J. & Gompper, G. Emergence of metachronal waves in cilia arrays. *Proc. Natl Acad. Sci. USA* **110**, 4470–4475 (2013).
50. Tam, D. & Hosoi, A. E. Optimal kinematics and morphologies for spermatozoa. *Phys. Rev. E* **83**, 045303 (2011).
51. Mettot, C. & Lauga, E. Energetics of synchronized states in three-dimensional beating flagella. *Phys. Rev. E* **84**, 061905 (2011).
52. Lauga, E. & Eloy, C. Shape of optimal active flagella. *J. Fluid Mech.* **730**, 2013.2370 (2013).
53. Lenaghan, S. C., Chen, J. & Zhang, M. J. Modeling and analysis of propulsion in the multiflagellated micoorganism Giardia lamblia. *Phys. Rev. E* **88**, 012726 (2013).
54. Chen, D. T. N., Heymann, M., Fraden, S., Nicastro, D. & Dogic, Z. ATP consumption of eukaryotic flagella measured at a single-cell level. *Biophys. J.* **109**, 2562–2573 (2015).
55. Klindt, G. S., Ruloff, C., Wagner, C. & Friedrich, B. M. Load response of the flagellar beat. *Phys. Rev. Lett.* **117**, 258101 (2016).
56. Ueno, H. et al. Mouse respiratory cilia with the asymmetric axonemal structure on sparsely distributed ciliary cells can generate overall directional flow. *Nanomed. Nanotechnol.* **8**, 1081–1087 (2012).
57. Bui, K. H., Yagi, T., Yamamoto, R., Kamiya, R. & Ishikawa, T. Polarity and asymmetry in the arrangement of dynein and related structures in the *Chlamydomonas* axoneme. *J. Cell Biol.* **198**, 913–925 (2012).
58. Wargo, M. J. & Smith, E. F. Asymmetry of the central apparatus defines the location of active microtubule sliding in *Chlamydomonas* flagella. *Proc. Natl Acad. Sci. USA* **100**, 137–142 (2003).
59. Geyer, V. F. Characterization of the flagellar beat of the single cell green alga *Chlamydomonas reinhardtii*. *Doctoral thesis* (2013).
60. Crenshaw, H. C. Kinematics of helical motion of microorganisms capable of motion with four degrees of freedom. *Biophys. J.* **56**, 1029–1035 (1989).
61. Crenshaw, H. C. & Edelstein-Keshet, L. Orientation by helical motion—II. Changing the direction of the axis of motion. *Bull. Math. Biol.* **55**, 213–230 (1993).
62. Friedrich, B. M. & Jülicher, F. Steering chiral swimmers along noisy helical paths. *Phys. Rev. Lett.* **103**, 068102 (2009).
63. Su, T. W., Xue, L. & Ozcan, A. High-throughput lensfree 3D tracking of human sperms reveals rare statistics of helical trajectories. *Proc. Natl Acad. Sci. USA* **109**, 16018–16022 (2012).
64. Bennett, R. R. & Golestanian, R. A steering mechanism for phototaxis in *Chlamydomonas*. *J. R. Soc. Interface* **12**, 20141164 (2015).
65. Shingyoji, C., Higuchi, H., Yoshimura, M., Katayama, E. & Yanagida, T. Dynein arms are oscillating force generators. *Nature* **393**, 711–714 (1998).
66. Kagami, O. & Kamiya, R. Translocation and rotation of microtubules caused by multiple species of *Chlamydomonas* inner-arm dynein. *J. Cell Sci.* **103**, 653–664 (1992).
67. Bayly, P. V. & Dutcher, S. K. Steady dynein forces induce flutter instability and propagating waves in mathematical models of flagella. *J. R. Soc. Interface* **13**, pii: 20160523 (2016).
68. Kamiya, R. & Okamoto, M. A mutant of *Chlamydomonas reinhardtii* that lacks the flagellar outer dynein arm but can swim. *J. Cell Sci.* **74**, 181–191 (1985).
69. Brokaw, C. J. & Kamiya, R. Bending patterns of *Chlamydomonas* flagella: IV. Mutants with defects in inner and outer dynein arms indicate differences in dynein arm function. *Cell Motil. Cytoskel.* **8**, 68–75 (1987).
70. Witman, G. B. *The Chlamydomonas Sourcebook: Cell Motility and Behavior.* Vol. 3 (Elsevier, 2009).

## Acknowledgements

Financial support by the Deutsche Forschungsgemeinschaft *via* the priority program SPP 1726 "Microswimmers" is gratefully acknowledged.

## Author contributions

J.E., L.A., U.B.K., and G.G. designed research; G.S. and J.E. performed data analysis and theoretical modeling; J.F.J. and L.A. performed experiment and image analysis; All authors discussed the results and wrote the paper.

## Additional information

**Competing interests:** The authors declare no competing financial interests.

