## [Peer Review File · Nature Communications]

Reviewers' comments:

Reviewer #1 (Remarks to the Author):

The authors report a new underlying mechanism of spermatozoa steering through studying flagella dynamics of human sperm cells tethered to surface. The manuscript reveals a correlation of the tethered spermatozoa rotation with the phase and the amplitude of the second harmonic in the flagella beat. The authors have also found that an addition of progesterone increases the rotation and decreases the beating frequency, suggesting that the change in calcium concentration correlates with an increase of the second harmonic contribution. The large amount of experimental data has been collected and analysed for describing the phenomena convincingly. The authors have derived an analytical approach, which breaks the flagella beating mirror symmetry and leads to the cell angular rotation through the intensity of the second harmonic. Furthermore, the authors conducted numerical simulations on active semi-flexible filament model, which agree with the analytical findings. Though the experimental data does not directly reveal the steering of the spermatozoa trajectories, the derived analytical model and the numerical simulations reproduce the suggested dynamics. The experimental findings appear to be consistent with the analytical and the numerical results for a 2D flagella stroke. The article is well written and easy to follow. The results elaborate on a potentially new mechanism of spermatozoa (and possibly other microswimmers) steering, which can be of an interest across multidisciplinary communities for understanding mechanisms of biological taxes for swimming cells.

The authors have mentioned previously reported rolling of the free swimming sperm cells. The rolling results in an effective 3D stroke through a rotation of the 2D beating plane. The importance of the 3D stroke can be seen in rheotaxis, which would otherwise be impossible for a 2D confined flagella dynamics. The tethering of the cells, as reported in the manuscript, suppresses the rolling, which would otherwise disturb the proposed steering mechanism by alternating the curvature in the case of a free swimming cell. I would appreciate if the authors provide a comment on the situation, where the cells would exhibit rolling while swimming.

A minor comment: The results on the energy dissipation are weakly connected to the main message of the manuscript and seem to stand alone, while it might be interesting to see a comparison for the stimulated and unstimulated cells.

Overall, I am in favour of the manuscript publication in Nature Communications as long as the authors address the above concerns.

Reviewer #2 (Remarks to the Author):

In this paper, the authors examine the beating of human sperm flagella and demonstrate that a second harmonic frequency is likely responsible for cells turning. Experimentally, they examine the beating modes of the flagellum to isolate a first and second harmonic in the beat frequencies, and hypothesize that the second harmonic is responsible for temporal symmetry breaking in the flagellar beating. They develop an analytical model that corroborates these observations and demonstrate how this symmetry breaking results in a torque that rotates the cell body. They also show that chemoattractants can amplify the presence of the second harmonic and exacerbate cell turning. Finally, integrate the idea of a second harmonic into a dynamic (elastic) model of the beating flagellum and examine the resulting energy consumption of the simulated swimming cells.

Overall, I found this to be a well written, accessible, and convincing study of human sperm motility. This work adds a new perspective to a vast body of work on the subject with obvious implications for human reproduction and cell motility. The quality and level of the paper are appropriate for Nature Communications, and I recommend it for publication. However, I have a few (mostly minor) comments and concerns listed below.

- Line 46: The authors state that the flagellum acts as a sensor for flow and temperature without providing citations. I feel that these sentiments are mostly conjectures in the field and have not seen convincing evidence otherwise. The authors should provide reference or tone down these statements.
- Line 64: What does "narrow" mean in this context?
- Line 66: One of my main criticisms of the work is in the implications of the experimental technique of tethering the cells to the substrate surface. In particular, this means that the cells are no longer force free or torque free entities, and that the cell is in very close proximity to the solid substrate (enhancing the drag force). While this tethering method has been used previously to study flagellar mechanics, the authors do not discuss the potential implications of these forces, and how they might change the interpretation of their results. Playing the devil's advocate: Is the second harmonic present in free-swimming cells? Did the authors attempt to examine this?
- Line 67: Regarding reference [8], I do not believe that this reference addresses the issue of planar vs. 3D beating in sperm near surfaces. As the authors likely know, planar beating and non-rotation of the sperm body are two different notions: Which precisely do the authors mean here? The flagellar beat of human sperm is slightly helicoidal. In this context how might a 3D beat modify the interpretation of the experiments due to variable friction with distance from the surface? There have been recent works which examine the changes in flagellar beating with distance from the surface - I believe that this is different from the effect to which the authors refer. In particular, Nosrati et al (Nat Comm, 2015) did indeed show that a subpopulation sperm swimming near a planar substrate exhibit a highly planar flagellar beat, which has vastly different kinematics from free swimming cells slightly above the surface. Again, how do the authors rectify their results in the context of this work?
- Line 174: This statement is slightly contradictory to the authors' main message. By the authors' own admission, the mean curvature effects do contribute to the cell rotation (albeit less than the second harmonic effects). Consider revising this message throughout.
- Line 178: Clarify what is meant by "hydrodynamic interactions."
- Figure 2d: The probability density data in this plot appears to be smoothed. This smoothing should be removed or the smoothing algorithm should be described. (it is possible that this is due to my local display, and in that case this comment may be disregarded)

Reviewer #3 (Remarks to the Author):

This study investigates the mechanism by which the direction of sperm swimming is controlled. The authors postulate that motion at the 2nd harmonic of the flagellar beat frequency is responsible for generating an asymmetric beat and net torque on the head, leading to a curved path. This is an alternative hypothesis to the idea that the curved swimming trajectory is caused by a static curved shape superimposed on the periodic flagella beat (as proposed in Geyer et al., 2016; Eshel and Brokaw, Cytoskeleton 7(2):160-168, 1987). This is a new and interesting idea.

STRENGTHS

The authors provide thorough and compelling experimental evidence that (1) sperm flagella curvature exhibits a strong component (quantified by amplitude C_2) at twice the beat frequency, and (2) that the amplitude of this 2nd harmonic correlates with rotation about the head.

They show analytically that a waveform with a significant 2nd harmonic of curvature is asymmetric

and generates torque. They further show by simulation that driving a flagella with distributed, periodic torques containing only 1st and 2nd harmonics leads to a non-zero average curvature of the flagellum and a curved trajectory.

These observations and calculations all appear to be valid. The combination of experiment, analysis, and simulation provides evidence on several fronts for the plausibility of the authors' ideas. The paper is well organized and clearly written.

WEAKNESSES

1. The authors do not clearly establish that the 2nd harmonic component of curvature variation (quantified by C_2) is more important than the static component of curvature (quantified by C_0). The authors do show that (i) the amplitude of C_2 is greater than C_0 , and (ii) that C_2 produces a contribution to net torque and angular rotation velocity (Eq. 4). They do not yet show what the corresponding effect of C_0 would be. They need an equation analogous to Eq. 4, showing the effect of C_0 , and they should compare the effect of average curvature C_0 to the effect of 2nd harmonic C_2 .

2. The authors do not establish causality: i.e., that 2nd harmonic temporal variations of motor activity cause static curvature and asymmetry of motion. They do show by simulations that this is possible. However, it is equally plausible that pre-curvature of the flagellum (an asymmetric, nonlinear system) excited by simple, single harmonic activity, can lead to 2nd harmonics in the response. Thus, 2nd harmonics in the waveform curvature can be the result, not the cause, of average curvature asymmetry. As an analogy, consider the oscillations of the simple nonlinear oscillator with asymmetric, nonlinear stiffness (Eq. R1):

$$m\ddot{x} + c\dot{x} + k_1 x + k_2 x^2 = f_0 + f_1 \sin(\omega t) \quad [R1]$$

As either the amplitude of excitation or the strength of the nonlinearity increase in Eq. R1, the second harmonic component of the response will increase.

The authors' model is akin to Eq. R2.

$$m\ddot{x} + c\dot{x} + k_1 x + k_2 x^2 = f_1 \sin(\omega t) + f_2 \sin(2\omega t) \quad [R2]$$

The point is that in a nonlinear system, the 2nd harmonic term on the right in Eq. R2 is not necessary in order to have 2nd harmonics in the response.

3. The authors do not propose, or refer to, any satisfactory molecular mechanism for generating 2nd harmonics at twice the beat frequency. I.e., what could produce the assumed active torques in Eq. 5? The current authors do refer to studies in *Chlamydomonas* that show some motors might prefer to work at half the beat frequency, but half the beat frequency is much different than twice the beat frequency. The lack of proposed mechanism for generating 2nd harmonics is in contrast to simple mechanisms of asymmetric average dynein activity that might cause non-zero average curvature (Geyer et al., 2016; Eshel and Brokaw, 1987).

There are papers out there that could be cited to support the authors' idea. A couple of recent studies (Bayly and Dutcher *J Roy Soc Interface*, 2016, Bayly and Wilson, *J Roy Soc Interface*, 2015) support the notion that higher-frequency modes might lose stability and contribute to higher-harmonic components of curvature, as average levels of dynein activity increase or other parameters change.

4. (minor) The authors use "small amplitude" (linear) approximations in their analysis, but the amplitudes of sperm motion are actually quite large (nonlinear). They should comment on the range of applicability of the "small-amplitude" model results, and interpret them accordingly.

We thank the reviewers for their constructive remarks and their positive criticism. In the following, we address their questions point-by-point:

Reviewer #1:

1. The authors have mentioned previously reported rolling of the free swimming sperm cells. The rolling results in an effective 3D stroke through a rotation of the 2D beating plane. The importance of the 3D stroke can be seen in rheotaxis, which would otherwise be impossible for a 2D confined flagella dynamics. The tethering of the cells, as reported in the manuscript, suppresses the rolling, which would otherwise disturb the proposed steering mechanism by alternating the curvature in the case of a free swimming cell. I would appreciate if the authors provide a comment on the situation, where the cells would exhibit rolling while swimming.

A cell propelled by regularly beating its flagellum with two harmonics has a period that is equal to that of the fundamental period. During this time, the net cell movement can be described by a translation and a rotation. For free-swimming sperm, the swimming path that results from a periodic translation and rotation in 3D is a helix or a twisted ribbon for a helix radius near zero (Crenshaw, HC Biophysical. J. 1989; Crenshaw, H and Edelstein-Keshet, L. 1993 Bulletin of Mathematical Biology; Friedrich and Jülicher Phys. Rev. Lett. 2009; Jikeli et al. Nature Comm. 2015). Here, we show using numerical simulations (Fig. 4e) that the phase between the two harmonics in 2D sets the curvature of the swimming path. For a constant phase, sperm would swim on a circular path with no net drift. For a net drift to occur, it is required that sperm adjust the phase between the two harmonics in time. In 3D, a constant phase between the two harmonics will result in a helix with a straight helical axis. For navigation in 3D (while sperm also roll), and in analogy with the 2D scenario, the phase between the two harmonics needs as well to be adjusted in time. Thus, the proposed mechanism for steering should still be valid, even in the 3D scenario. We extended the discussion to address this point.

New Main Text:

“Steering by a second-harmonic mechanism should be also valid for rolling sperm moving freely in three dimensions (3D). The 3D swimming path resulting from a non-planar periodic beat is a helix or a twisted ribbon for a helix radius near zero^{22,63-65}. Analogous to the 2D scenario, for steering in 3D the phase between the two harmonics needs to be adjusted as well. Thus, the proposed second-harmonic mechanism might be also valid for 3D swimming. However, studies of human sperm swimming in 3D reveal that only a small fraction of sperm (4-5%) swims on a regular helical path⁶⁶. Future work needs to establish the role of the second harmonic for steering in 3D when the swimming path deviates from a perfect helix.”

2. A minor comment: The results on the energy dissipation are weakly connected to the main message of the manuscript and seem to stand alone, while it might be interesting to see a comparison for the stimulated and unstimulated cells.

We agree that comparing stimulated and unstimulated cell is interesting. We presented such a comparison in the Supplementary Information of the original manuscript, but this was weakly connected to the corresponding Main Text. We now explicitly mention the comparison between unstimulated and stimulated sperm in the Main Text and refer to the corresponding Supplementary figure:

“The dissipated power remains unaltered after stimulation (Supplementary Fig. 8)”

We also include a more detailed comparison between these cases in the Supplementary Notes:

“For both unstimulated and stimulated sperm, a simple linear fit yields

$$P_d(s) = P_d^0 + P_d^1 s, \quad (\text{S20})$$

where $P_d^0 = 0.265 \pm 0.101$ [fW/ μm] and $P_d^1 = 0.023 \pm 0.008$ [fW/ μm^2] for unstimulated sperm and $P_d^0 = 0.253 \pm 0.075$ [fW/ μm] and $P_d^1 = 0.022 \pm 0.007$ [fW/ μm^2] after stimulation. Thus, dissipated power does not change upon stimulation.”

We considered incorporating this information and the corresponding Supplementary Fig. 8 into the Main Text, but we felt that this detail does not reveal additional insight and would be better suited for the Supplementary Information.

Reviewer #2:

1. *Line 46: The authors state that the flagellum acts as a sensor for flow and temperature without providing citations. I feel that these sentiments are mostly conjectures in the field and have not seen convincing evidence otherwise. The authors should provide reference or tone down these statements.*

We completely agree with the referee that many of these claims have not been sufficiently substantiated and the field, in fact, is characterized by sentiments and conjectures without rigorous proof. We now rephrase this paragraph and provide references.

New Main Text:

“It has been proposed that the flagellum also serves as antenna that registers sensory cues as diverse as chemoattractant molecules¹²⁻¹⁴, fluid flow^{15,16}, or temperature¹⁷⁻¹⁹. The sensory cues modify the flagellar beat pattern and, thereby, guide sperm to the egg^{20,21}. Whereas chemotaxis, the directed movement in a chemical gradient, has been firmly established in

sperm from marine invertebrates and plants²¹⁻²⁷, it is debated which sensory cues guide mammalian, in particular human sperm to the egg^{12-14,28}. Whichever mechanisms might underlie steering, it is commonly accepted that changes in intracellular Ca^{2+} concentration ($[\text{Ca}^{2+}]_i$) modulate the flagellar beat and, thereby, swimming direction^{20,29-32}."

2. Line 64: What does "narrow" mean in this context?

We specified the dimensions of the recording chamber explicitly in the Material and Methods section. For convenience, we include it in the Main Text.

3. Line 66: *One of my main criticisms of the work is in the implications of the experimental technique of tethering the cells to the substrate surface. In particular, this means that the cells are no longer force free or torque free entities, and that the cell is in very close proximity to the solid substrate (enhancing the drag force). While this tethering method has been used previously to study flagellar mechanics, the authors do not discuss the potential implications of these forces, and how they might change the interpretation of their results. Playing the devil's advocate: Is the second harmonic present in free-swimming cells? Did the authors attempt to examine this?*

The reviewer's comment is well taken. We presume that tethered sperm are indeed not force-free cells, but that they can still be approximated as torque free: The torque generated by the beat is compensated by cell rotation. We are currently establishing holographic techniques to study the flagellar beat in 3D, which will be required to further characterize steering of freely swimming sperm. Please note, however, that as we state in the manuscript, the second harmonic has been previously noted in freely swimming sperm from bull (Friedrich, B. et al., J. Exp. Biol. 2010), and reactivated flagella from *Chlamydomonas* (Geyer, V., PhD Thesis 2013) but has not been considered as a means of flagellar steering. These studies argue that the second harmonic does not result from cell tethering. We present these arguments in the discussion of the revised version.

New Main Text:

"It might be argued that the second harmonics of the beat result from the physical constraints imposed by cell tethering. However, second harmonics have also been noted in freely swimming sperm from bull⁶, and in reactivated flagella from *Chlamydomonas*⁶². Moreover, simulations of the flagellar beat show that the second-harmonic intensity can control both the path curvature of freely swimming sperm and the rotation velocity of tethered sperm (Fig. 4e, Supplementary Movie 2). Together, these observations indicate that the second-harmonic mechanism *per se* does not result from tethering constraints."

4. *Line 67: Regarding reference [8], I do not believe that this reference addresses the issue of planar vs. 3D beating in sperm near surfaces. As the authors likely know, planar beating and non-rotation of the sperm body are two different notions: Which precisely do the authors mean here?*

We thank the referee for spotting this source of potential misunderstanding. Indeed, planarity and rolling are intertwined. We rephrase the sentence accordingly. Nonetheless, please note that Bukatin et al. (PNAS 2015) quantify the planarity ratio P of the flagellar beat. They find a P of about 0.2, indicating that the flagellar beat is fairly planar.

New Main Text:

“We monitored the flagellar beat of human sperm in a shallow recording chamber (150 μm depth) filled with an aqueous buffer solution (viscosity $\sim 0.7 \text{ mPa}\cdot\text{s}$ at 37°C) (Fig. 1a). While swimming near a surface at low viscosities ($< 20 \text{ mPa}\cdot\text{s}$), human sperm undergo a rolling motion^{8,33}. Rolling occurs despite the fact that the flagellar beat is almost planar⁸. We prevent cell rolling by tethering sperm with their head to the recording chamber. Under these conditions, the beat plane remains parallel to the surface, which facilitates tracking and imaging of the flagellar motion.”

5. *The flagellar beat of human sperm is slightly helicoidal. In this context how might a 3D beat modify the interpretation of the experiments due to variable friction with distance from the surface? There have been recent works which examine the changes in flagellar beating with distance from the surface - I believe that this is different from the effect to which the authors refer. In particular, Nosrati et al (Nat. Comm., 2015) did indeed show that a subpopulation sperm swimming near a planar substrate exhibit a highly planar flagellar beat, which has vastly different kinematics from free swimming cells slightly above the surface. Again, how do the authors rectify their results in the context of this work?*

Nosrati et al. (Nature Comm. 2015) examine the movement of bull and human sperm near surfaces. In that study, 39% of bull sperm swimming within 4 μm from the surface display a flagellar beat that is almost planar (confined within 1 μm height). However, to observe a similar planar beat for human sperm, high viscosities are required (above 20 $\text{mPa}\cdot\text{s}$). The effect of such higher viscosities were not explored in our study. We specify the viscosity of the media used during our experiments and refer to the suggested study from Nosrati et al. We also extended the discussion as to encourage future studies on freely moving cells.

6. *Line 174: This statement is slightly contradictory to the authors' main message. By the authors' own admission, the mean curvature effects do contribute to the cell rotation (albeit less than the second harmonic effects). Consider revising this message throughout.*

Indeed, the title of this subsection might seem contradictory. Nonetheless, we predict that the presence of the second harmonic results in a small average curvature. This small average

curvature, however, cannot explain the observed rotation velocity. See also answer to referee 3, question 1. We, therefore, chose another caption for this subsection, because there are additional more important insights than just the one previously highlighted.

New title of the subsection:

“An active elastic-filament model predicts that sperm could navigate by adjusting the phase between the two harmonics.”

7. Line 178: Clarify what is meant by “hydrodynamic interactions.”

The local friction experienced by a particle moving in a fluid depends on the relative velocity of the fluid flow. In turn, fluid flow is affected by the movement of the particle itself, i.e., fluid flow and particle motion are coupled. In a similar vein, in the presence of multiple particles, individual particles are coupled to the fluid, implying that the motions of particles are affected by each other via the fluid. Such interactions are referred to as hydrodynamic interactions. For a sperm flagellum, different parts of the flagellum interact with each other while moving in the fluid. For a slender body, such as the sperm flagellum, hydrodynamic interactions can be approximated by an anisotropic local friction tensor: The friction for pulling a thin filament perpendicular to its long axis is roughly twice as large as for pulling along this axis (Gray, J. & Hancock, G. J. *J. Exp. Biol.* 1955; Friedrich, B. et al *J. Exp. Biol.* 2010; Lighthill, J. Flagellar Hydrodynamics. *SIAM Rev.* 18, 161–230 (1976)). This is the basis of resistive-force theory of sperm motion (Gray, J. & Hancock, G. J. *J. Exp. Biol.* 1955).

We have briefly extended the respective sentence for clarity:

“A sperm cell is modelled as an actively beating filament of bending rigidity κ ; local hydrodynamic interactions resulting from the dynamic coupling between different portions of the flagellum via the induced fluid flow are taken into account by anisotropic drag^{6,37,39}.”

8. Figure 2d: The probability density data in this plot appears to be smoothed. This smoothing should be removed or the smoothing algorithm should be described. (it is possible that this is due to my local display, and in that case this comment may be disregarded)

We thank the referee for spotting this shortcoming. Due to figure format conversion, the probability density appears smoothed. We replaced the figure accordingly.

Reviewer #3:

1. The authors do not clearly establish that the 2nd harmonic component of curvature variation (quantified by C2) is more important than the static component of curvature

(quantified by C_0). The authors do show that (i) the amplitude of C_2 is greater than C_0 , and (ii) that C_2 produces a contribution to net torque and angular rotation velocity (Eq. 4). They do not yet show what the corresponding effect of C_0 would be. They need an equation analogous to Eq. 4, showing the effect of C_0 , and they should compare the effect of average curvature C_0 to the effect of 2nd harmonic C_2 .

This is indeed an important point. We agree with the referee that a more explicit formulation of an equation analogous to Eq. 4 facilitates the assessment of the contribution of each amplitude. We included this equation in the Main Text (new Eq. 5). As mentioned in the Main Text, for equal magnitudes of C_0 and C_2 , both mechanisms contribute almost equally to the rotation frequency. However, the average intrinsic curvature of the flagellum is usually much smaller than the amplitude of the second harmonic ($|C_0|/|C_2|=0.13$; Supplementary Notes and Supplementary Fig. 5). Therefore, we conclude that the second-harmonic contribution to rotation velocity dominates.

2. The authors do not establish causality: i.e., that 2nd harmonic temporal variations of motor activity cause static curvature and asymmetry of motion. They do show by simulations that this is possible. However, it is equally plausible that pre-curvature of the flagellum (an asymmetric, nonlinear system) excited by simple, single harmonic activity, can lead to 2nd harmonics in the response. Thus, 2nd harmonics in the waveform curvature can be the result, not the cause, of average curvature asymmetry. As an analogy, consider the oscillations of the simple nonlinear oscillator with asymmetric, nonlinear stiffness (Eq. R1):

$$m\ddot{x} + c\dot{x} + k_1 x + k_2 x^2 = f_0 + f_1 \sin(\omega t) \quad [R1]$$

As either the amplitude of excitation or the strength of the nonlinearity increase in Eq. R1, the second harmonic component of the response will increase.

The authors' model is akin to Eq. R2.

$$m\ddot{x} + c\dot{x} + k_1 x + k_2 x^2 = f_1 \sin(\omega t) + f_2 \sin(2\omega t) \quad [R2]$$

The point is that in a nonlinear system, the 2nd harmonic term on the right in Eq. R2 is not necessary in order to have 2nd harmonics in the response."

This is also an important issue. We agree with the referee that higher harmonics of the beat frequency can also result from nonlinearities. Two different sources on nonlinearities can be identified: structural (mechanical properties of complex materials) and geometric nonlinearities (resulting from body deformation). The simulations presented in the original manuscript take into consideration geometric nonlinearities. Inspired by the referee's comment, we have performed additional simulations to investigate whether second harmonics result from geometric nonlinearities, even in the absence of second-harmonic driving torques (Eq. 6). Specifically, we have simulated sperm featuring an oscillating torque

at the fundamental frequency and a constant torque component that results in an average curvature. This constant torque has been set such that the resulting average curvature

- a) matches that from experiments, or
- b) is large enough to produce the rotation frequency observed from experiments

Indeed, geometric nonlinearities and a constant torque result in a spontaneous second harmonic frequency. However, when taking an average curvature observed in experiments (case a), the simulated flagellum does not rotate as fast as tethered sperm. For simulations with either an average curvature set to match experimental rotation frequency (case b) or second-harmonic torque, beat shape and flagellar rotation are similar. Nevertheless, a more quantitative analysis reveals that in case b) the mean amplitude ratio ($\langle C_0/C_2 \rangle \approx 6$) is much larger than that obtained in the second-harmonic-torque simulations ($\langle C_0/C_2 \rangle \approx 0.5$), and is even larger than the experimental ratio ($\langle C_0/C_2 \rangle \approx 0.13$). Thus, we conclude that second-harmonic torques provide the best fit to the experimental results. It might be possible to consider other forms of nonlinearity. However, to be useful, a detailed understanding of the mechanisms underlying second-harmonic generation is required. The elucidation of the origin of the second harmonic (or even the first one) goes far beyond the scope of our manuscript.

In the revised version, we include an additional movie (Supplementary Movie 3) comparing the second-harmonic simulation (blue; see Eq. (6) and Supplementary Movie 1) with the scenarios (a) (orange) and (b) (black). In addition, a comment concerning the intrinsic curvature and elastic nonlinearities has been added to the Main Text and is further discussed in the Supplementary Information.

New SI Text:

“Second harmonics of the beat could be produced from the superposition of fundamental and second harmonic waves of active bending torques as described in the Main Text (see Eq. (6)). Alternatively, second harmonics could result from elastic nonlinearities. We tested this hypothesis by using our simulation approach. Specifically, we simulated sperm cells lacking a second harmonic torque, but featuring a constant torque component that results in an average curvature (Supplementary Movie 3). This constant torque has been set such that the resulting average curvature

- (1) matches that from experiments, or
- (2) is large enough to produce the cell rotation frequency observed from experiments.

Indeed, including nonlinearities results in a spontaneous second-harmonic frequency. However, when taking an average curvature observed in experiments (case 1), the simulated flagellum does not rotate as fast as tethered sperm. For simulations with either an average curvature set to match experimental rotation frequency (case 2) or a second-harmonic

torque, beat shape and flagellar rotation are similar. Nevertheless, a more quantitative analysis reveals that in case 2) the mean amplitude ratio ($\langle C_0/C_2 \rangle \approx 6$) is much larger than that obtained in the second-harmonic-torque simulations ($\langle C_0/C_2 \rangle \approx 0.5$), and is even larger than the experimental ratio ($\langle C_0/C_2 \rangle \approx 0.13$). Thus, we conclude that second-harmonic torques provide the best fit to the experimental results. It might be possible to consider other forms of nonlinearity. However, to be useful, a detailed understanding of the mechanisms underlying second-harmonic generation is required. The elucidation of the origin of the second harmonic (or even the first one) goes far beyond the scope of our manuscript.”

New manuscript text:

“Finally, two different sources of flagellar nonlinearities are possible: structural nonlinearities resulting from anharmonic mechanical properties of complex materials and geometric nonlinearities resulting from body deformation. Nonlinearities could also produce a spectrum of flagellar beat frequencies, particularly first and second harmonics⁷⁰. We investigated this hypothesis by simulations of elastic filaments including geometric nonlinearities and a bending torque with a fundamental component superimposed upon an average torque, but no second harmonic component (see Supplementary Information). Indeed, nonlinearities result in a spontaneous second-harmonic frequency. However, a bending torque with fundamental and second-harmonic components (Eq. 6) clearly provides the best fit to the experimental results.”

3. The authors do not propose, or refer to, any satisfactory molecular mechanism for generating 2nd harmonics at twice the beat frequency. I.e., what could produce the assumed active torques in Eq. 5?

Indeed, the mechanism for second-harmonic generation is unknown. As a matter of fact (see for example Bayly and Dutcher 2016), it is even unclear how the first harmonic is generated, even though this very question is at the focus of research for many years. It is beyond the scope of this study to provide the molecular mechanism underlying second-harmonic generation. However, in the discussion section we discuss studies hinting to molecular components that might be involved.

4. The current authors do refer to studies in Chlamydomonas that show some motors might prefer to work at half the beat frequency, but half the beat frequency is much different than twice the beat frequency. The lack of proposed mechanism for generating 2nd harmonics is in contrast to simple mechanisms of asymmetric average dynein activity that might cause non-zero average curvature (Geyer et al., 2016; Eshel and Brokaw, 1987). There are papers out there that could be cited to support the authors' idea. A couple of recent studies (Bayly and Dutcher J Roy Soc Interface, 2016, Bayly and Wilson, J Roy Soc Interface, 2015) support the notion that higher-frequency modes might lose stability and contribute to higher-harmonic components of curvature, as average levels of dynein activity increase or other parameters change.

The referee raises an interesting issue. As suggested in the original manuscript, the frequencies observed for mutants lacking either the outer or the inner dynein arms is different by a factor of about two. This suggests that the different dynein arms are tuned to produce different beat frequencies. We thank the referee for pointing out the recent studies by Bayly et al. We extended the discussion accordingly and included the suggested literature.

Extended Main Text discussion:

“The mechanisms underlying the second harmonic and its modulation by Ca^{2+} are not known. Here, we discuss some potential mechanisms. Second harmonic generation might involve dynein motors. Dynein arms behave as endogenous oscillators that slide microtubules with a frequency set by the ATP concentration⁶⁸. In fact, isolated *Chlamydomonas* flagella that were reactivated with different ATP concentrations display beat amplitudes with two peak resonances at 30 and 60 Hz⁶². For a given ATP concentration, different dynein isoforms translocate microtubules at different velocities⁶⁹. Thus, fundamental and second harmonics could be produced by different motor subsets. A novel model for the generation of flagellar waves predicts that a combination of steady forces, i.e. dynein activity, and dynamic instability can create oscillatory waves propagating down the flagellum. The beat frequency does not rely on switching or modulation of motor activity, but on the force density produced by motors on the flagellum⁷⁰. In fact, detergent-extracted axonemes from sea urchin sperm and flagella from *Chlamydomonas* mutants lacking the outer dynein arms beat at about half the frequency⁷¹⁻⁷³, supporting the notion that a lower motor activity or density results in reduced beat frequency⁷⁰. By contrast, *Chlamydomonas* flagella with defective inner dynein arms beat at about the same frequency as wild type flagella, suggesting that inner and outer dynein arms produce different beat frequencies^{72,73}. Finally, two different sources of flagellar nonlinearities are possible: structural nonlinearities resulting from anharmonic mechanical properties of complex materials and geometric nonlinearities resulting from body deformation. Nonlinearities could also produce a spectrum of flagellar beat frequencies, particularly first and second harmonics⁷⁰. We investigated this hypothesis by simulations of elastic filaments including geometric nonlinearities and a bending torque with a fundamental component superimposed upon an average torque, but no second harmonic component (see Supplementary Information). Indeed, nonlinearities result in a spontaneous second-harmonic frequency. However, a bending torque with fundamental and second-harmonic components (Eq. 6) clearly provides the best fit to the experimental results. Future studies need to address the mechanisms by which a second-harmonic mode is created and tuned for steering.”

5. (minor) *The authors use “small amplitude” (linear) approximations in their analysis, but the amplitudes of sperm motion are actually quite large (nonlinear). They should comment on the range of applicability of the “small-amplitude” model results, and interpret them accordingly.*

The validity range can only be estimated by considering explicitly higher-order terms, and the beat amplitude at which they become significant. This would require a systematic expansion of the full model in powers of the beat amplitude. We have chosen a different approach:

- We employ the small-amplitude approximation to demonstrate analytically the existence of a torque induced by the second-harmonic amplitude.
- We perform a small-curvature approximation to complement the small-amplitude calculation; this approximation remains valid for larger beat amplitudes.
- We perform simulations of the full model, which includes all higher-order terms.

The comparison of the results of these three approaches indicates that the small-amplitude calculation provides the proper scaling. We changed the Main Text accordingly:

“For comparison with experiments, it is more useful to describe the waveform $y(x,t)$ in terms of local curvature $C(s,t)$, with amplitudes C_1 and C_2 instead of y_1 and y_2 in equation (2). In addition, a small-curvature approximation is more accurate for larger beat amplitudes.”

“Third, simulations confirm two predictions from the small-curvature approximation Eq. (4): The rotation velocity Ω scales linearly both with T_2 (Fig. 4C and D) and with the sine of the phase ψ (Fig. 4D and S7).”

REVIEWERS' COMMENTS:

Reviewer #1 (Remarks to the Author):

The authors have addressed previously raised concerns. Now, I can recommend the manuscript for publication in Nature Communications.

Reviewer #2 (Remarks to the Author):

The authors have adequately address my concerns and I believe those of the other reviewers. I am still in favor that this work should be published in Nature Communications.

Reviewer #3 (Remarks to the Author):

The reviewers have adequately addressed the questions raised in the first review.